# Did an urban perinatal health programme in Rotterdam, the Netherlands, reduce adverse perinatal outcomes? Register-based retrospective cohort study

Hendrik CC de Jonge,[1] Jacqueline Lagendijk  ,[2] Unnati Saha,[1] Jasper V Been,[1,2,3] Alex Burdorf[1]

¹Department of Public Health, Erasmus MC, Rotterdam, The Netherlands
²Department of Obstetrics and Gynaecology, Erasmus MC, Rotterdam, The Netherlands
³Department of Paediatrics, Erasmus University Medical Centre–Sophia Children's Hospital, Rotterdam, The Netherlands

**Correspondence to**
Hendrik CC de Jonge;
h.c.c.dejonge@erasmusmc.nl

## ABSTRACT

**Objectives** To study the effect of an urban perinatal health programme in Rotterdam, the Netherlands, on perinatal outcomes.

**Design** A retrospective cohort study with difference-in-differences analysis using individual-level perinatal outcome data from the Dutch Perinatal Registry 2003–2014 linked to Central Bureau of Statistics data of migration background and individual disposable household income.

**Intervention** The programme consisted of perinatal health promotion, risk selection and risk-guided pregnancy care, and a new primary care child birth centre. The programme was implemented during 2009–2012.

**Primary outcome measures** We compared trends in perinatal mortality, preterm delivery and small-for-gestational-age births between targeted urban neighbourhoods in Rotterdam (n=61 415) and all other urban neighbourhoods in the Netherlands (n=881 202). The effect of the programme was modelled as a change in trend of each perinatal outcome in the treatment group post intervention compared with the control population from January 2010 onwards. All analyses were adjusted for maternal age, parity, ethnicity and individual-level low socioeconomic status (SES). We also conducted a stratified analysis by SES.

**Results** During 2003–2014, downward trends in perinatal mortality (adjusted OR (aOR) 0.9439 per year, 95% CI 0.9362 to 0.9517), preterm birth (aOR 0.9970 per year, 95% CI 0.9944 to 0.9997) and small-for-gestational-age births (aOR 0.9809 per year, 95% CI 0.9787 to 0.9831) in the entire study population were observed. No demonstrable changes in these trends were found in the intervention group after the programme had started. The stratified analyses by SES showed no changes in trends post intervention in both strata either.

**Conclusions** The programme had no demonstrable effects on perinatal outcomes. The intervention may not have reached a sufficient proportion of the population or has provided too little contrast to the widespread attention for inequalities in pregnancy outcomes occurring simultaneously in the Netherlands.

## Strengths and limitations of this study

► The nationally representative sample of almost a million pregnancies during the period 2003–2014 had a good discriminatory power to estimate the effect of the programme on a rare outcome like perinatal mortality.

► The linkage across three routinely collected data registries provided pregnancy outcomes and socioeconomic data, allowing adjustment for important medical and social determinants in the statistical analysis.

► We applied difference-in-differences analysis, an econometric method, that allows us to determine programme effects on population level.

► All pregnant women in the urban neighbourhoods targeted with the intervention were considered to be exposed to the intervention, whether or not these women were actually reached by activities within the complex intervention, which may have attenuated observed associations.

► The relative good performance of the urban perinatal healthcare system in Rotterdam before the intervention period and unobserved improvements in urban perinatal healthcare during the intervention period in the Netherlands may have attenuated the observed intervention effect.

## INTRODUCTION

In the 2000s, the Netherlands had a relatively high perinatal mortality, ranking third highest among 26 European countries in 2004 and sixth highest in 2010.[1–3] Considerable regional inequalities in perinatal outcomes were reported within the Netherlands.[4] In Rotterdam, the second largest city in the Netherlands, perinatal mortality was markedly higher than nationally, most notably in its deprived neighbourhoods.[5] In a childbirth cohort study, these neighbourhood differences in various perinatal outcomes

could largely be attributed to the increased prevalence of medical, as well as, social risk factors.[6]

In response to these findings, the Erasmus MC in collaboration with the Municipal Health Service Rotterdam-Rijnmond initiated an urban perinatal health programme, called 'Ready for a Baby', with the aim to improve perinatal outcomes in Rotterdam. The programme consisted of several intervention components across the pregnancy care chain: preconception health promotion, improved risk selection and risk-guided care during pregnancy, and the establishment of a primary care birth centre (PCBC). These components were gradually introduced in the period 2009–2012, and, depending on the component, reached nearly city-wide coverage (preconception health promotion) or only specific neighbourhoods (eg, PCBC).[7]

Perinatal mortality in the Netherlands has gradually reduced over the past two decades. Favourable trends in risk factors may have contributed, including a reduction in smoking by pregnant women, less multiple births, increased use of ultrasonography at 20 weeks gestation for detection of congenital abnormalities and improved care for very premature babies.[8 9] The introduction of the Ready for a Baby programme in Rotterdam was conceptualised as a natural experiment.[10] In order to evaluate whether this programme has had an additional impact on the secular trends of decline in unfavourable perinatal outcomes, the difference-in-differences (DiD) method was considered the appropriate approach to evaluate the effects of an intervention in an observational study. In this method, the change in health in the intervention group before and after the introduction of the intervention (difference) can be distinguished from changes in health over time in both the intervention and control groups (differences).[11] Therefore, the aim of this study was to determine the influence of the urban perinatal health programme Ready for a Baby on adverse perinatal health outcomes.

## METHODS

### Study design and population

We conducted a retrospective cohort study on routinely collected birth data from the Dutch Perinatal registry, enriched with national registers with personal information to evaluate the influence of a community intervention with a DiD analysis. The start in year 2003 was determined by individual information on socioeconomic status (SES) becoming available in national registers. The year 2014 was the last year with complete information available at time of this study. The study population consisted of all singleton deliveries in urban neighbourhoods in the Netherlands during the study period, comprising approximately 45% of all deliveries in the Netherlands. A neighbourhood was defined as a postal code area and an urban neighbourhood as a postal code area with more than 1500 houses per square kilometre.[12] The intervention group consisted of all deliveries in

all 51 urban neighbourhoods in 10 out of 14 boroughs in Rotterdam, where at some point in time during the intervention period a component of the urban perinatal health programme was implemented. The control group comprised all deliveries in other urban postal code areas in the Netherlands, including six untargeted urban neighbourhoods in Rotterdam.

### Programme description

The Ready for a Baby programme had three components: health promotion in preconception care, improved risk selection and risk-guided care, and establishment of a PCBC in the university medical centre. The content of the programme has been published in detail before and will be described here briefly.[7]

The first component of the programme aimed to promote preconception health by three strategies. The first strategy aimed to collectively increase awareness of the importance of preconception health through mass media campaigns (including flyers, posters, editorials and advertisements in local Dutch and Turkish newspapers, on buses and trams, at offices of healthcare providers, pharmacists, retailers and at churches and mosques). Besides increasing awareness, these campaigns aimed to promote favourable attitudes and behaviours for a healthy pregnancy, such as the use of folic acid and the cessation of smoking and alcohol use. This strategy targeted all citizens in Rotterdam and was not confined to the intervention group. The second strategy used peer education to increase preconception-related health literacy and motivation to attend preconception care consultations, especially among low SES and migrant groups.[13] Peer educators recruited about 2300 participants during the course of the programme for peer education group sessions through their existing network and community meeting places (eg, mosques and schools). The sessions were interactive, often in multiple languages, and provided participants information on the influence of lifestyle changes on pregnancy outcomes, and additionally advised on where to obtain individual consultations. The third strategy in targeted neighbourhoods was the provision of individual preconception care consultations by general practitioners (GPs) and community midwives. At least 43 couples attended an individual consultation after receiving tailored health promotion from a web-based preconception health assessment 'Preconceptiewijzer.nl'.[13 14]

The second component of the programme aimed to improve risk selection and risk-guided care by use of the Rotterdam Reproductive Risk Reduction (R4U) scorecard along with the Shared Care model[15–17] in the targeted neighbourhoods. The R4U scorecard is a systematic risk assessment in the first trimester of pregnancy focusing on medical and non-medical risk factors related to adverse pregnancy outcomes, including, for example, migration background and low household income.[16 17] The Shared Care model is an approach to risk-guided care that has three elements: (1) continuity of care

(eg, a case manager is assigned to high-risk women who need care from different professionals; care pathways are defined for risks identified in the R4U), (2) patient centeredness (eg, through fostering of self-management, and efforts to combine appointments to different care providers), and (3) interprofessional collaboration (eg, through formulating a joint set of aims and ambitions for collaboration including care pathways, training in team work and interprofessional education).[15] The R4U scorecard guided the care pathways in the Shared Care model through templates describing the consecutive steps a professional was advised to take to reduce the potential contribution of identified risk factors for adverse perinatal health outcomes. In order to enhance the efficiency and quality of the local antenatal healthcare, all details of instruments and templates were discussed during meetings with community midwives, obstetricians and social workers under guidance of a member of the Ready for a Baby programme. Since midwives are usually the first point of contact of a pregnant woman, the vast majority (n=46) of all community midwives in the intervention area were trained to use and integrate the R4U scorecard along with the Shared Care model into their daily practice. In three selected boroughs, the programme supported the use of care pathways in midwifery practices and hospitals with multidisciplinary meetings to follow-up on high-risk cases.

The third component of the programme was the establishment of a PCBC in the Erasmus University Medical Centre in Rotterdam. The PCBC is a separate facility led by community midwives, where women can deliver at their own discretion if a hospital delivery is not medically indicated but social factors make a home delivery less preferable.[18 19] The PCBC aimed to provide risk-directed care by assessing the risk status of each woman at different intervals (eg, on arrival at the PCBC, during labour and postpartum).

Figure 1 shows how the different parts of the intervention were introduced from July 2009 until December 2012. The first component of the programme, preconception health promotion, targeted primarily women in selected neighbourhoods, but by nature of the instrument the

mass media approach had a reach in all neighbourhoods in Rotterdam. The second component, the risk selection and shared care, was implemented in the selected neighbourhoods. The third component, the new birth centre, primarily served pregnant women from the community midwife practices around Erasmus MC. So, the different components were introduced at different moments in time and in different neighbourhoods.

## Data

We used data from the Dutch Perinatal Registry 2003–2014 (Perined, https://www.perined.nl). The Dutch Perinatal Registry is a registry with information provided by midwives, gynaecologists, paediatricians and GPs and contains demographic characteristics, medical risk factors, obstetric history and pregnancy and neonatal outcomes.[20 21] Through individual-level linkage, the Dutch Perinatal Registry was enriched with two nationwide registries from Statistics Netherlands. Household income from the Integral Household Income register, based on tax information from 2003 onwards,[22] was used to define low SES by the lowest 20% disposable household income. The second, the Dutch population register, provided information on migration background.[23]

A trusted third party (an in-house service at Statistics Netherlands) merged these datasets using four-digit postal code, birth date of the mother and birth date of the child after which the identifying variables were removed. The merged one-way coded dataset was made available in a secure research environment at Statistics Netherlands for analysis. Results were rigorously checked for identifiability by Statistics the Netherlands before they were released from the secure research environment for publication. This procedure is in accordance with Dutch legislation and the Dutch Code of Conduct for Medical Research for use of anonymous data for research purposes without an explicit informed consent.[24]

Deliveries with less than 24 completed weeks of gestation from the dataset were excluded, because most of these deliveries will end in stillbirth. These stillbirths are not registered in the population registry according to Dutch law so cannot be linked to household income and migration background data, which were required for the statistical analysis. Records with other missing covariates for the statistical analysis were also excluded.

## Outcomes

Our primary outcomes were: perinatal mortality, preterm birth and small-for-gestational-age (SGA) birth. Perinatal mortality was defined as the number of stillbirths from 24 completed weeks of gestation or early neonatal deaths (ie, death of a liveborn baby within the first 7 days of life) per 1000 singleton births. Preterm birth was defined as birth before 37 completed weeks of gestation per 1000 singleton livebirths. SGA was defined as any baby that was smaller than the 10th percentile, corrected for gestational age in weeks and sex, according to the Visser curve, per 1000 singleton livebirths.[25]

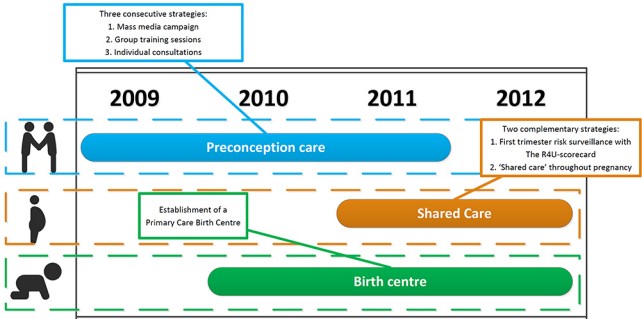

**Figure 1** A graph showing the different parts of the intervention 'Ready for a Baby' that were introduced from July 2009 until December 2012. R4U, Rotterdam Reproductive Risk Reduction.

## Statistical analysis

A DiD analysis was conducted with logistic regression models for perinatal mortality, preterm birth and SGA as dichotomous dependent variables.

Each model has three key independent variables and several confounders. The first variable is the year of birth as continuous variable, which captures the trend over time and is expressed by the OR for trend per year that represents the yearly increase or decrease in likelihood of perinatal outcome. The second variable is a dummy variable with value 1 for the intervention group and value 0 for the control group. This dummy variable reflects differences in perinatal health between intervention and control groups at baseline. The third variable is an interaction term between years since start of the programme as continuous variable and intervention group status. This term corresponds to the DiD estimate, as it presents differences in perinatal health trends between intervention and control groups post intervention over and above the underlying temporal trends.[15] It is important to note that this DiD estimate presents the change in slope of the trend for the intervention group post intervention. Thus, we did not model the invention as an immediate effect (step function) that introduces a constant difference between intervention and control group from start of the programme until 2014. The programme was gradually introduced from July 2009 onwards and, given the duration of pregnancy, we assumed that the intervention could take effect first from 2010 onwards. Therefore, a DiD analysis on change in the trend per year in the intervention group after 2010 was considered most appropriate.

A crucial assumption in the DiD analysis is the parallel trend assumption,[15] that is, that preintervention trends were similar in intervention and control groups over the period 2003–2009. This assumption was assessed with graphs of the perinatal outcomes per year 2003–2009 (online supplementary files 1-3). The assumption was also tested with a regression model on the preintervention period 2003–2009 with an interaction between intervention and a dummy variable for year of birth 2003–2009, which indicates whether the baseline difference between intervention and control group changed per year. The graphs and regression model showed that the parallel trend assumption was not violated for any of the perinatal outcomes (online supplementary file 4).

We included four major risk factors that were targeted in the intervention (SES, ethnicity, parity and age) as confounders. This step was taken to improve the comparability of the intervention and control groups in the analysis. SES is associated with perinatal outcomes and was differently distributed in the intervention and control groups; therefore, the lowest 20 percentile household income was used as indicator of SES. As the intervention targeted particularly low-SES women, we also conducted stratified analyses according to SES. Ethnicity is associated with perinatal outcomes and with urbanisation, and hence was included as confounder. Non-Dutch ethnicity was defined as any person who was born in another country than the Netherlands (first generation migration background) or had at least one parent born in another country (second generation migration background). We also included parity and age at delivery as potential confounders.

We conducted several sensitivity analyses for lagged programme effects, adjustment for covariate imbalance and the individual programme component effects. Lagged programme effects were studied with a regression model with an interaction of the intervention and a dummy variable for each year in the period 2010–2014. Adjustment for covariate imbalance in our main analysis was handled by ordinary regression analysis, which is appropriate given the number of observations.[26] Propensity score matching as alternative approach was conducted as a sensitivity analysis.[27] For the propensity score model, we used the same set of variables for matching as in the main analysis (age, parity, migration background, household income). Matching was done per year of delivery using the nearest neighbour algorithm. We evaluated the balance as sufficient by inspecting a table with the distribution of the outcomes and covariates. As a final sensitivity analysis, we studied a possible intervention effect in the boroughs that were targeted by these interventions, compared with the control population, using the same DiD model as in the main analysis.

### Patient and public involvement

This research was done without patient or public involvement. However, the research question was proposed by the Municipal Health Service Rotterdam as part of their legal task of monitoring population health (Dutch Public Health Law).

## RESULTS

### Study population

From 2003 to 2014, a total of 2 116 226 deliveries were available in the Dutch Perinatal Registry, encompassing 96.6% of all deliveries in the Netherlands in that period [20–21]. A total of 1 031 683 deliveries were registered within an urban neighbourhood of the intervention and control group, thus representing 49% of all deliveries in the Netherlands (figure 2). We excluded all deliveries before 24 weeks of completed gestation and all multiple births, which were 38 931 deliveries (3.8% of all deliveries) of which 3704 perinatal deaths (39% of all perinatal deaths (online supplementary file 5)). In addition, we excluded deliveries that could not be matched to Statistics Netherlands data (household income or migration background) (16 319 deliveries, 1.6%) or lacked information on any other confounder (33 819 deliveries, 3.3%). Thus, the total study population comprised 942 614 deliveries (figure 2).

In the intervention group 83% of all deliveries took place in deprived urban neighbourhoods, whereas in the control group the corresponding figure was 39% (table 1).

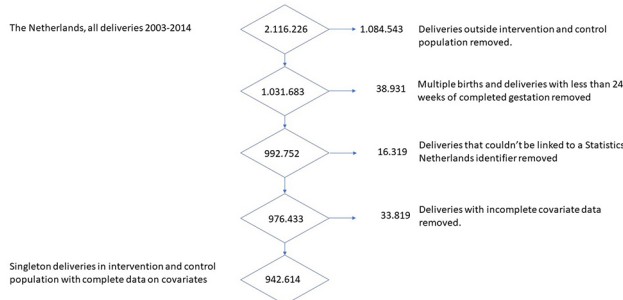

**Figure 2** A flow chart displaying how the study population was generated from the Perined data for the Netherlands, 2003–2014.

Women delivering in the intervention group were more often younger, multiparous, non-Dutch and more often had a household income below the 20th percentile. The percentage of women with more than one of these risk factor was considerably higher in the intervention population (59%) than in the control population (34%). In addition, in deprived neighbourhoods in the intervention group, pregnant women had a higher prevalence of risk factors than pregnant women in deprived neighbourhoods of the control group, specifically non-Dutch background (73% vs 49%) and low dispensable household income (46% vs 35%). The incidence rates for perinatal mortality, preterm birth and for SGA were higher in the intervention group than in the control group, both in deprived as well as non-deprived neighbourhoods.

### Difference-in-differences analysis

Table 2 shows that the likelihood of perinatal mortality decreased during 2003–2014 by about 6% per year across all urban neighbourhoods in the Netherlands (adjusted OR (aOR) 0.9439/year (95% CI 0.9362 to 0.9517)). SGA decreased by 2% per year (aOR 0.9809 (95% CI 0.9787 to 0.9831)) and preterm birth decreased by 0.3% per year (aOR 0.9970 (95% CI 0.9944 to 0.9997)). Small differences in annual trends between the intervention and the control group were observed, although not statistically significant. The DiD analysis on the effect of the urban perinatal health programme showed that during the postintervention period in the intervention group perinatal mortality (aOR 1.0535 (95% CI 0.9889 to 1.1223) increased slightly each year, whereas preterm birth (aOR 0.9809 (95% CI 0.9619 to 1.0004)) and SGA (aOR 0.9928 (95% CI 0.9772 to 1.0086)) showed modest improvements.

The analysis showed that for the preintervention period, after adjustment for important confounders, perinatal mortality was 14% lower in the intervention population than in the control population (aOR 0.8601, 95% CI 0.7534 to 0.9819). Preterm birth (aOR 1.0962 (95% CI 1.0505 to 1.1440)) and SGA (aOR 1.1313 (95% CI 1.0935 to 1.1703)) were higher in the intervention population than in the control population.

The stratified analysis by SES of the pregnant woman showed very similar results for pregnant women with low

SES and pregnant women with higher SES. The intervention had no demonstrable influence on trends in any of the perinatal outcomes post intervention (table 3).

The sensitivity analysis for lagged programme effects did not indicate any lagged programme effect (online supplementary file 6). The sensitivity analysis using propensity score matching gave similar results to the main analysis (online supplementary file 7). No intervention effects were observed for individual programme components (online supplementary file 8).

### DISCUSSION

In this study, the influence of an urban perinatal health programme in Rotterdam on perinatal health outcomes was evaluated by a DiD approach, an analytical technique for natural experiment evaluation in an observational setting. The DiD analysis could not demonstrate that the introduction of the programme influenced trends in perinatal mortality, preterm birth or SGA birth in the postintervention years in the intervention group.

Strengths of this study include the large study population and the available information on important confounders. The nationally representative sample of almost a million pregnancies during the period 2003–2014 had a good discriminatory power to estimate the effect of the programme on a rare outcome like perinatal mortality. The linkage across three routinely collected data registries provided pregnancy outcomes and socio-economic data, allowing adjustment for important medical and social determinants in the statistical analysis. The modelling was checked for robustness by several sensitivity analyses.

Our study also has several limitations pertaining to data availability, and to the DiD analysis. Availability of data was dictated by the registers used. SES is an important determinant of perinatal health,[28] but in our analysis only disposable household income was available. Highest educational attainment based on certified diploma registers had a high percentage of missing values, and could therefore not be included as a covariate in our study. Using only a one-dimensional representation of SES might not fully adjust for residual differences between the intervention and the control group.

The DiD analysis applied in this study has several limitations. First, a crucial assumption is that trends in outcome in intervention and control groups in the years before the intervention are parallel, that is, have a constant difference, captured in the DiD logistic regression model by the difference at baseline (table 2). Evaluation showed that this assumption was not violated. The DiD analysis accounts for time-invariant differences between the intervention and control groups, as well as any factors that equally change over time in both groups. The descriptive information showed that in the intervention neighbourhoods in Rotterdam there is much higher accumulation of risk factors among pregnant women, and that a much larger proportion of women

**Table 1** Sociodemographic characteristics, presence of risk factors, and perinatal mortality, preterm birth, and small-for-gestational age over the period 2003–2014 in intervention group (targeted urban neighbourhoods in Rotterdam) and control group (other urban neighbourhoods in the Netherlands), stratified by deprivation of postal code area*

| | Intervention area | | | Control areas | | |
|---|---|---|---|---|---|---|
| | All | Deprived | Non-deprived | All | Deprived | Non-deprived |
| Total no of deliveries | 61 415 | 50 727 | 10 688 | 881 202 | 340 914 | 540 288 |
| **Parity** | | | | | | |
| Nulliparous | 48% / 29 308 | 46% / 23 180 | 57% / 6128 | 47% / 418 074 | 47% / 160 270 | 48% / 257 804 |
| 1–2 | 44% / 26 880 | 45% / 22 716 | 39% / 4164 | 47% / 417 538 | 46% / 156 892 | 48% / 260 646 |
| 3+ | 9% / 5227 | 10% / 4831 | 4% / 396 | 5% / 45 590 | 7% / 23 752 | 4% / 21 838 |
| Ethnicity (non-*Dutch*) | 68% / 41 702 | 73% / 36 951 | 44% / 4751 | 37% / 321 950 | 49% / 167 943 | 29% / 154 007 |
| **Age of mother** | | | | | | |
| <25 | 20% / 12 423 | 22% / 11 319 | 10% / 1104 | 13% / 110 465 | 18% / 60 191 | 9% / 50 274 |
| 25–29 | 30% / 18 361 | 31% / 15 679 | 25% / 2682 | 28% / 250 322 | 31% / 105 834 | 27% / 144 488 |
| 30–34 | 31% / 18 917 | 29% / 14 522 | 41% / 4395 | 37% / 327 361 | 33% / 111 349 | 40% / 216 012 |
| 35–39 | 16% / 9616 | 15% / 7491 | 20% / 2125 | 19% / 164 403 | 16% / 53 252 | 21% / 111 151 |
| ≥40 | 3% / 2098 | 3% / 1716 | 4% / 382 | 3% / 28 651 | 3% / 10 288 | 3% / 18 363 |
| **Socioeconomic status** | | | | | | |
| Lowest 20% disposable household income | 42% / 25 919 | 46% / 23 431 | 23% / 2488 | 24% / 212 405 | 35% / 120 488 | 17% / 91 917 |
| More than 1 risk factor† | 59% / 36 139 | 63% / 29 638 | 43% / 6501 | 34% / 299 727 | 49% / 129 418 | 27% / 170 309 |
| **Perinatal mortality (/1000 stillbirths and live singleton births)** | | | | | | |
| 2003–2009 | 6.5 / 223 | 6.7 / 202 | 4.8 / 21 | 6.2 / 3173 | 7.1 / 1436 | 5.7 / 1737 |
| 2010–2014 | 5.1 / 136 | 5.7 / 116 | 3.2 / 20 | 4.5 / 1667 | 5.2 / 725 | 4.0 / 942 |
| **Preterm birth rate (/1000 live singleton births)** | | | | | | |
| 2003–2009 | 63 / 2166 | 64 / 1917 | 57 / 249 | 56 / 28 447 | 60 / 12 066 | 54 / 16 381 |
| 2010–2014 | 60 / 1611 | 61 / 1254 | 57 / 357 | 55 / 20 376 | 59 / 8098 | 53 / 12 278 |
| **SGA rate (1000 live singleton births)** | | | | | | |
| 2003–2009 | 109 / 3756 | 113 / 3377 | 87 / 379 | 84 / 42 505 | 96 / 19 330 | 76 / 23 175 |
| 2010–2014 | 94 / 2513 | 98 / 2007 | 80 / 506 | 76 / 28 249 | 86 / 11 898 | 70 / 16 351 |

*Deprivation is defined as the lowest quintile of postal codes by status score, an area level composite index of individual household income, education and employment status calculated every 4 years by the Netherlands Institute of Social Research.
†The following risk factors are considered: nulliparous, non-Dutch ethnicity, age of the mother <25 year, lowest 20% disposable household income.
SGA, small-for-gestational age.

de Jonge HCC, *et al. BMJ Open* 2019;9:e031357. doi:10.1136/bmjopen-2019-031357

**Table 2** Logistic regression models with difference-in-differences analyses of the effect of the urban perinatal health programme on perinatal mortality, preterm birth and small-for-gestational age (SGA)

| Independent variables | Perinatal mortality | | Preterm birth | | SGA | |
| --- | --- | --- | --- | --- | --- | --- |
| | OR | 95% CI | OR | 95% CI | OR | 95% CI |
| Trend per year | 0.9439 | 0.9362 to 0.9517 | 0.9970 | 0.9944 to 0.9997 | 0.9809 | 0.9787 to 0.9831 |
| Difference intervention and control group | 0.8601 | 0.7534 to 0.9819 | 1.0962 | 1.0505 to 1.1440 | 1.1313 | 1.0935 to 1.1703 |
| Programme effect (change in trend per year from 2010 onwards) | 1.0535 | 0.9889 to 1.1223 | 0.9809 | 0.9619 to 1.0004 | 0.9928 | 0.9772 to 1.0086 |
| Low SES | 1.3761 | 1.2917 to 1.4660 | 1.2017 | 1.1764 to 1.2275 | 1.4417 | 1.4170 to 1.4668 |
| Ethnicity non-Dutch | 1.2690 | 1.3462 to 1.1963 | 1.0068 | 1.0264 to 0.9876 | 1.2927 | 1.3134 to 1.2722 |
| Parity 0 (reference) | 1.0000 | | 1.0000 | | 1.0000 | |
| Parity 1–2 | 0.7560 | 0.7120 to 0.8026 | 0.6114 | 0.5998 to 0.6232 | 0.5340 | 0.5254 to 0.5427 |
| Parity 3+ | 1.2823 | 1.1541 to 1.4248 | 0.8218 | 0.7894 to 0.8554 | 0.4871 | 0.4687 to 0.5062 |
| Age <25 (reference) | 1.0000 | | 1.0000 | | 1.0000 | |
| Age 25–34 | 0.9401 | 0.8654 to 1.0212 | 0.9786 | 0.9529 to 1.0050 | 0.8554 | 0.8377 to 0.8735 |
| Age≥35 | 1.2238 | 1.1112 to 1.3479 | 1.0762 | 1.0423 to 1.1111 | 0.9339 | 0.9100 to 0.9585 |
| Baseline rate* | 8.6 | 7.8 to 9.4 | 71 | 69 to 74 | 155 | 151 to 159 |

Perinatal mortality is defined as stillbirth from 24 weeks onwards plus early neonatal mortality. The total number of observations for the perinatal mortality model is 942 614. Preterm is defined as born before a gestational age of 37 weeks and SGA is defined as a birth weight below the 10th percentile corrected for gestational age and sex. The total number of observations for the preterm birth and SGA models is 937 639.
*For perinatal mortality per 1000 live and stillbirths, for preterm birth and SGA per 1000 live births.
SES, socioeconomic status.

lived in deprived neighbourhoods. In our analysis, we adjusted for these risk factors to ensure comparability of intervention and control groups. However, it cannot be discarded that women in Rotterdam have experienced less favourable trends in other, unobserved, risk factors during the observation period of this study, which may have attenuated any potential effect of the intervention. Second, the DiD makes the strong assumption that the precise timing of the intervention is completely at random, creating exogenous variation that allows causal inference. This assumption cannot be formally tested, but it must be acknowledged that the urban perinatal health programme was designed and implemented in response to the relatively high perinatal mortality in the Netherlands and opportunities for improvement in prevention and child healthcare in Rotterdam.

At the start of the Ready for a Baby programme, adjusted for risk factors, perinatal mortality was lower in Rotterdam than in other urban areas in the Netherlands. This favourable position of Rotterdam might be partly attributed to the large concentration of hospitals in Rotterdam and attention in the local child healthcare system for high-risk women since they constitute a relatively large part of all pregnant women. Therefore, the DiD analysis may have not been able to capture additional change in an already decreasing trend in perinatal mortality in the intervention group. A linked issue is that improvements in perinatal healthcare may have occurred also in other neighbourhoods. These cointerventions may have biased the comparisons between intervention and control groups.

It is important to consider that this study evaluates the possible influence of a population intervention rather than the effects of an intervention at individual level. The content of the complex intervention comprises universal primary prevention and changes in quality and delivery of child healthcare, which are notoriously difficult to evaluate at individual level. Also, the different components of the intervention were introduced gradually during the intervention period and had varying coverage: from nearly city wide to certain neighbourhoods only. In the analysis, all pregnant women in the targeted urban neighbourhoods with the intervention were considered to be exposed to the intervention, whether or not these women were actually reached by activities within the complex intervention. This might have attenuated observed associations, when a substantial number of women would not have been included in components of the programme. Only 43 couples attended an individual preconception consultation by a GP in all of Rotterdam, which is too small to expect any effect on population level. In contrast, the majority of midwifes was trained to use the R4U scorecard and Shared Care method, resulting in a large uptake of the second component of the programme. However, we lack information on compliance of the implementation of the R4U and Shared Care in daily practice.

A fair question to ask is whether the DiD evaluation of the programme Ready for a Baby sufficiently reflects the underlying improvements. The programme had several interacting components of universal and high-risk prevention embedded in improvements in quality of child healthcare and can therefore be characterised as a complex

**Table 3** Stratified analysis by socioeconomic status (SES) of the mother with logistic regression models with difference-in-differences analyses of the effect of the urban perinatal health programme on perinatal mortality, preterm birth and small-for-gestational age (SGA)

| Lowest quintile household income | Perinatal mortality | | Preterm birth | | SGA | |
|---|---|---|---|---|---|---|
| **Independent variables** | **OR** | **95% CI** | **OR** | **95% CI** | **OR** | **95% CI** |
| Trend per year | 0.9516 | 0.9380 to 0.9654 | 1.0008 | 0.9958 to 1.0059 | 0.9864 | 0.9825 to 0.9903 |
| Difference intervention and control group | 0.8209 | 0.6786 to 0.9929 | 1.1677 | 1.0957 to 1.2444 | 1.1296 | 1.0749 to 1.1871 |
| Programme effect (change in trend per year from 2010 onwards) | 1.0673 | 0.9761 to 1.1670 | 0.9767 | 0.9481 to 1.0062 | 0.9947 | 0.9718 to 1.0181 |
| Ethnicity non-Dutch | 1.3668 | 1.5181 to 1.2307 | 0.8716 | 0.9024 to 0.8418 | 1.0167 | 1.0450 to 0.9893 |
| Parity 0 (reference) | 1.0000 | | 1.0000 | | 1.0000 | |
| Parity 1–2 | 0.7392 | 0.6652 to 0.8214 | 0.7826 | 0.7545 to 0.8117 | 0.6258 | 0.6084 to 0.6438 |
| Parity 3 | 1.0544 | 0.8960 to 1.2407 | 1.0134 | 0.9539 to 1.0766 | 0.5605 | 0.5308 to 0.5918 |
| Age <25 (reference) | 1.0000 | | 1.0000 | | 1.0000 | |
| Age 25–34 | 1.0666 | 0.9478 to 1.2004 | 0.9566 | 0.9188 to 0.9960 | 0.8914 | 0.8647 to 0.9190 |
| Age≥35 | 1.2893 | 1.1082 to 1.4999 | 1.1172 | 1.0597 to 1.1778 | 0.9334 | 0.8947 to 0.9737 |
| Baseline rate* | 11.2 | 9.8 to 12.7 | 70 | 67 to 74 | 181 | 175 to 188 |
| Household income above 20th percentile | Perinatal mortality | | Preterm birth | | SGA | |
| **Independent variables** | **OR** | **95% CI** | **OR** | **95% CI** | **OR** | **95% CI** |
| Trend per year | 0.9402 | 0.9308 to 0.9497 | 0.9961 | 0.9930 to 0.9992 | 0.9787 | 0.9761 to 0.9813 |
| Difference intervention and control group | 0.9084 | 0.7553 to 1.0926 | 1.0472 | 0.9884 to 1.1094 | 1.1500 | 1.0978 to 1.2046 |
| Intervention effect (change in trend per year from 2010 onwards) | 1.0294 | 0.9403 to 1.1270 | 0.9841 | 0.9585 to 1.0103 | 0.9876 | 0.9663 to 1.0093 |
| Ethnicity non-Dutch | 1.2108 | 1.3017 to 1.1262 | 1.0665 | 1.0913 to 1.0423 | 1.4399 | 1.4678 to 1.4126 |
| Parity 0 (reference) | 1.0000 | | 1.0000 | | 1.0000 | |
| Parity 1–2 | 0.7592 | 0.7057 to 0.8166 | 0.5592 | 0.5467 to 0.5721 | 0.4956 | 0.4859 to 0.5056 |
| Parity 3+ | 1.4987 | 1.3058 to 1.7201 | 0.7298 | 0.6905 to 0.7713 | 0.4446 | 0.4206 to 0.4700 |
| Age <25 (reference) | 1.0000 | | 1.0000 | | 1.0000 | |
| Age 25–34 | 0.8417 | 0.7489 to 0.9461 | 0.9544 | 0.9206 to 0.9894 | 0.8185 | 0.7950 to 0.8427 |
| Age≥35 | 1.1253 | 0.9880 to 1.2816 | 1.0446 | 1.0024 to 1.0886 | 0.9142 | 0.8835 to 0.9459 |
| Baseline rate* | 9.2 | 8.1 to 10.4 | 80 | 76 to 83 | 179 | 173 to 185 |

Perinatal mortality is defined as stillbirth from 24 weeks onwards plus early neonatal mortality. The total number of observations for the perinatal mortality model is 238 226 for low SES and 704 388 for high SES. Preterm is defined as born before a gestational age of 37 weeks and SGA is defined as a birth weight below the 10th percentile corrected for gestational age and sex. The total number of observations for the preterm birth and SGA models is 236 737 for low SES and 701 524 for high SES. Low SES was defined as the lowest 20 percentile by disposable household income per year.
*For perinatal mortality per 1000 live and stillbirths, for preterm birth and SGA per 1000 live births.

intervention. Complex interventions usually develop in phases from series of pilots to fully scaled up programme and should preferably be tested using a phased approach, starting with a series of pilot studies and moving on to an exploratory and then a definitive evaluation.[29] The programme Ready for a Baby could be described as the first phase in the development of a programme to reduce inequalities in perinatal outcomes and the lessons learnt from the programme were included in the next phase, the 'Healthy Pregnancy 4 All' programme.[30] Therefore, the results of the evaluation of the first programme Ready for a Baby should be interpreted with care. Other studies and other research methods are needed to better understand the underlying mechanisms of reach, uptake and effectiveness of specific programme activities (eg, action research to understand the dynamics of the developing pilots).

In conclusion, in this DiD analysis, we could not demonstrate an influence of the urban perinatal health Ready for a Baby on perinatal outcomes. Epidemiological evidence of inequalities in adverse pregnancy outcomes is compelling enough to justify continued efforts to develop a healthcare system that can properly deal with social risk factors. It is advised to evaluate such a system when it has been brought to scale and matured sufficiently to have a discernible impact.[31] The Ready for a Baby programme

generated a lot of attention both locally and nationally for the relevance of social determinants of pregnancy outcomes and for the development of methods to integrate obstetric care and the social domain, which is a valuable outcome in itself.

**Acknowledgements** The authors would like to acknowledge Statistics the Netherlands and the Netherlands Perinatal Registry for their support with the data. They would like to thank Yannan Hu and Marti Rado for their methodological support and Semiha Denktas for critically reviewing the manuscript. This research project was conducted as part of the academic collaborative "Centre for Effective Public Health In the larger Rotterdam area".

**Contributors** HCCdJ, JVB and AB were involved in the conception and design of the study. HCCdJ and JVB acquired the data from the Netherlands Perinatal Registry and Statistics the Netherlands. HCCdJ and US analysed the data and all authors were involved with the interpretation of the results. HCCdJ and JL drafted the original work and all authors were involved in revising it critically. All authors have approved the final version of the manuscript.

**Funding** JVB is supported by a personal fellowship from the Netherlands Lung Foundation.

**Competing interests** None declared.

**Patient consent for publication** Not required.

**Provenance and peer review** Not commissioned; externally peer reviewed.

**Data availability statement** Results are based on calculations by Erasmus MC using non-public microdata from Statistics Netherlands. Under certain conditions, these microdata are accessible for statistical and scientific research. For further information: cvb@cbs.nl. Syntax files that allow repeating the analyses in this paper from microdata at Statistics the Netherlands can be obtained from the first author.

**ORCID iD**
Jacqueline Lagendijk http://orcid.org/0000-0002-7178-8910

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
