## [Reviewer comments · BMJ Open]

ARTICLE DETAILS

TITLE (PROVISIONAL)	Did an urban perinatal health programme in Rotterdam, the Netherlands, reduce adverse perinatal outcomes? A register-based retrospective cohort study.
AUTHORS	de Jonge, Hendrik C. C.; Lagendijk, Jacqueline; Saha, Unnati; Been, JV; Burdorf, Alex

VERSION 1 – REVIEW

REVIEWER	Xu Ji CDC, U.S.A
REVIEW RETURNED	16-Jun-2019

GENERAL COMMENTS	This paper innovatively examined the effect of an urban prenatal health program on perinatal outcomes using a difference-in-difference (DiD) framework. This manuscript is carefully thought out, and I particularly appreciate the clarity in writing and the description of program details. My main comments are about the setup of the DiD model, which may influence the validity of the null effect observed. These and other concerns are outlined below. 1. It takes time for a program to fully come into effect and have an impact on health outcomes, and also given the time needed between pregnancy and delivery, it is not surprising to see a null program impact on pregnancy outcomes. However, have the authors looked at the impact of the program on the changes in intermediate outcomes, particularly maternal behaviors (e.g., use of prenatal vitamin, maternal smoking, maternal drinking) and health care utilization (e.g., adequate access to prenatal care)? Based on the program mechanisms described on pages 5-6, these intermediate outcomes are the direct target of the program.2. Because components of the program were gradually introduced during 2009-2012, it might be important to test both the average and the lagged policy effects. To do so, the authors may consider two alternative DiD model specifications. In one, use an indicator for “post” (i.e., coded as “1” for years 2010 and after, and as “0” for earlier years), and the average DiD estimate was the coefficient of the interaction term between “post” and “intervention group.” In another, include year indicators (a dummy variable for each year in 2010 and after) and also interact in each year indicator with “intervention group” to test the lagged policy effects (i.e., how program effects progress over time after implementation). There might be a lagged program effect on outcomes in later years post-implementation.3. The authors may consider excluding data for the calendar year of 2009, or at least for the period from July 2009 to December 2009, in
---

	order to minimize the influence of the transition period. 4. On page 8, the author described the pre-program parallel trend assumption tests. It would be helpful if the authors present the results from the tests at least in the appendix, as it is a crucial component for a valid DiD analysis. It might be better to also visually present the descriptive trends of the outcomes of interest before and after the program implementation in the intervention and the control groups. Such visualization will provide a clearer picture of what the pre-program parallel trend looks like, as well as how the program effects progressed over time post-implementation. 5. Page 9 and Table 1 showed the substantial differences between the intervention group and the control group with regard to some key sample characteristics. Given these difference, it might be helpful to apply the propensity score weighting or matching method, in combination with the DiD framework, to help to cope with these differences. This can be a sensitivity analysis as a robustness check of the findings. Other potential methods to handle differences between the intervention and comparison groups may be the synthetic control method. 6. The authors may consider conducting DiD falsification tests (e.g., pick a “fake” program start date) during the pre-implementation period, as a robustness check of the findings. 7. One argument for the null program effects is the “unobserved improvements in urban perinatal health care during the intervention period” which may confound the observed program effect. Is there any way to create measurements for these activities ongoing during the study period (which influenced the outcomes of study) using data available and control them as additional covariates in the DiD models?
--	--

REVIEWER	Hilary Brown University of Toronto
REVIEW RETURNED	17-Jun-2019

GENERAL COMMENTS	This study describes the impact of an urban perinatal health program in Rotterdam, Netherlands, on perinatal mortality, preterm birth, and small for gestational age. The study uses a difference-in-difference approach to test this natural experiment. The intervention includes many interesting and timely elements, including a focus on preconception health. The data are well-described, and the analysis is well-designed. My concerns relate to the design of the intervention itself and how suitable it is to evaluation using the authors’ methods. Major points:  - The authors state that the perinatal programme was introduced gradually across 2009 to 2012 (vs. a specific start date) and that not all of the components reached the entire city (e.g., primary care birth centres were only in certain neighbourhoods). Could the choice of looking at all (but 6) urban neighbourhoods in Rotterdam (vs. the specific neighbourhoods in which all of the components were implemented) explain why no effect was seen? It seems to me that the study population for this paper does not exactly match where the intervention was implemented, and therefore there is a mismatch between the study question and how it was tested. Could the authors identify the neighbourhoods in which all elements of the program were implemented and compare these to the ones in which none were implemented?
---

	- If the intervention was gradually introduced in 2009 to 2012, 2014 may not be a sufficient length of follow-up – i.e., depending on the component of the program there were only 2-5 years of post-intervention data. Why did the authors not analyze more recent data (e.g., up to 2018)? - Only 43 couples attended an individual consultant session in all of Rotterdam. Although 2,300 individuals received peer counseling, the number of people who received the strongest “dose” of preconception care was very small. This should be considered more fully in the Discussion section. Minor points: - What are the medical and social risk factors responsible for the heightened adverse perinatal outcomes in deprived neighbourhoods in the Netherlands? An understanding of these factors would help to understand why the perinatal programme may not have made an impact (i.e., did it target these risk factors or something else?). - What were the R4U risk factors that were targeted?
--	---

VERSION 1 – AUTHOR RESPONSE

Reviewer(s)' Comments to Author:

Reviewer: 1

Reviewer Name: Xu Ji

Institution and Country: CDC, U.S.A

Please state any competing interests or state 'None declared': None.

Please leave your comments for the authors below

This paper innovatively examined the effect of an urban prenatal health program on perinatal outcomes using a difference-in-difference (DiD) framework. This manuscript is carefully thought out, and I particularly appreciate the clarity in writing and the description of program details. My main comments are about the setup of the DiD model, which may influence the validity of the null effect observed. These and other concerns are outlined below.

1. It takes time for a program to fully come into effect and have an impact on health outcomes, and also given the time needed between pregnancy and delivery, it is not surprising to see a null program impact on pregnancy outcomes. However, have the authors looked at the impact of the program on the changes in intermediate outcomes, particularly maternal behaviors (e.g., use of prenatal vitamin, maternal smoking, maternal drinking) and health care utilization (e.g., adequate access to prenatal care)? Based on the program mechanisms described on pages 5-6, these intermediate outcomes are the direct target of the program.

Response: We agree with the reviewer that it takes time for a program to come into effect and an analysis of intermediate outcomes would have been interesting. Unfortunately, our primary data source, the Dutch Perinatal Registry, contains limited data on maternal behaviour and health care utilization. Maternal smoking is available but the variable is far from complete and the truthfulness is questionable. Maternal alcohol consumption is not available. Limited data on prenatal care is available (date of first visit). Therefore we focussed on an evaluation of health outcomes.

2. Because components of the program were gradually introduced during 2009-2012, it might be important to test both the average and the lagged policy effects. To do so, the authors may consider two alternative DiD model specifications. In one, use an indicator for “post” (i.e., coded as “1” for years 2010 and after, and as “0” for earlier years), and the average DiD estimate was the coefficient of the interaction term between “post” and “intervention group.” In another, include year indicators (a dummy variable for each year in 2010 and after) and also interact in each year indicator with

“intervention group” to test the lagged policy effects (i.e., how program effects progress over time after implementation). There might be a lagged program effect on outcomes in later years post-implementation.

Response: We ran both models suggested by the referent during the preparation of our paper. The first specification gave an average DiD estimate close to zero with no statistical significance, comparable to our main analysis. The second specification, whereby we introduced lagged effects, did not show a lagged policy effect (supplementary file 6). Given our overall conclusion, no effects of the programme, the results of these additional models are fully in line with our main findings.

Sentence in our revised methods (page 8, paragraph 4):

“We conducted several sensitivity analyses for lagged programme effects, adjustment for covariate imbalance and the individual program component effects. Lagged programme effects were studied with a regression model with an interaction of the intervention and a dummy variable for each year in the period 2010-2014.”

Sentence in our revised results (page 14, paragraph 2):

“The sensitivity analysis for lagged programme effects did not indicate any lagged programme effect (supplementary file 6).”

3. The authors may consider excluding data for the calendar year of 2009, or at least for the period from July 2009 to December 2009, in order to minimize the influence of the transition period.

Response: we appreciate the suggestion of the reviewer: a transition period could influence the results of our analysis. We think a transition period could be the first half of 2010, given the first program activities started in July 2009 and the time between intervention and effect is 22 weeks or more (minimal gestation to be registered in the Dutch Perinatal Registry). We ran the proposed analysis, which had similar results compared to the main analysis.

4. On page 8, the author described the pre-program parallel trend assumption tests. It would be helpful if the authors present the results from the tests at least in the appendix, as it is a crucial component for a valid DiD analysis. It might be better to also visually present the descriptive trends of the outcomes of interest before and after the program implementation in the intervention and the control groups. Such visualization will provide a clearer picture of what the pre-program parallel trend looks like, as well as how the program effects progressed over time post-implementation.

Response: The tests and graphs are provided as supplementary file 2-5. In our revised method section we now included a paragraph (presented below) addressing the parallel trend assumption in more detail (page 8, paragraph 2).

Paragraph in the revised methods (page 8).

“A crucial assumption in the DiD analysis is the parallel trend assumption (15), i.e. that pre-intervention trends were similar in intervention and control groups over the period 2003-2009. This assumption was assessed with graphs of the perinatal outcomes per year 2003-2009 (supplementary file 2-4). The assumption was also tested with a regression model on the pre-intervention period 2003-2009 with an interaction between intervention and a dummy variable for year of birth 2003-2009, which indicates whether the baseline difference between intervention and control group changed per year (supplementary file 5). The graphs and regression model showed that the parallel trend assumption was not violated for any of the perinatal outcomes.”

5. Page 9 and Table 1 showed the substantial differences between the intervention group and the control group with regard to some key sample characteristics. Given these difference, it might be helpful to apply the propensity score weighting or matching method, in combination with the DiD framework, to help to cope with these differences. This can be a sensitivity analysis as a robustness check of the findings. Other potential methods to handle differences between the intervention and comparison groups may be the synthetic control method.

Response: We thank the reviewer for this valuable comment. There are indeed considerable differences in key sample characteristics between the control and intervention group. We used regular logistic regression to adjust for these imbalances given that we had a considerable number of events per confounder (Soledad Cepeda, 2003). However, regression models may be subject to interpolation and extrapolation bias. To address this we conducted a robustness test namely, we applied a propensity score matching and ran the difference-in-difference model on the matched dataset (Stuart, 2010). We used the same set of variables for matching as in the main analysis (age, parity, migration background, household income). Matching was done per year of delivery using the nearest neighbor algorithm. We evaluated the balance as sufficient by inspecting a table with the distribution of the outcomes and covariates. The DiD model on the matched dataset produced similar results compared to the main analysis (supplementary file 7). We have added information regarding several sensitivity analyses to our revised method section (page 14, paragraph 2), as shown below.

Sentence in our revised methods (page 8, paragraph 4):

“We conducted several sensitivity analyses for lagged programme effects, adjustment for covariate imbalance and the individual program component effects.... Adjustment for covariate imbalance in our main analysis was handled by ordinary regression analysis, which is appropriate given the number of observations. Propensity score matching as alternative approach was conducted as a sensitivity analysis. For the propensity score model we used the same set of variables for matching as in the main analysis (age, parity, migration background, household income). Matching was done per year of delivery using the nearest neighbour algorithm. We evaluated the balance as sufficient by inspecting a table with the distribution of the outcomes and covariates.”

Sentence in our revised results (page 14, paragraph 2):

“The sensitivity analysis using propensity score matching gave similar results to the main analysis (supplementary file 7).”

6. The authors may consider conducting DiD falsification tests (e.g., pick a “fake” program start date) during the pre-implementation period, as a robustness check of the findings.

Response: We ran analyses for January 1st 2007, 2008 and 2009 as “fake” starting dates of the program with similar result to the main analysis.

7. One argument for the null program effects is the “unobserved improvements in urban perinatal health care during the intervention period” which may confound the observed program effect. Is there any way to create measurements for these activities ongoing during the study period (which influenced the outcomes of study) using data available and control them as additional covariates in the DiD models?

Response: We cannot identify these improvements, hence the “unobserved”, but Ready for a Baby started at a point in time where the Dutch perinatal health care system was about to change. Following the Peristat report in 2008 indicating a relatively high perinatal mortality in the Netherlands compared to other European countries, there was increased public attention for adverse perinatal outcomes. The 2010 report “A good start” by the ministerial committee “Pregnancy and birth care” recommended amongst other things better collaboration in pregnancy care, a public health approach for preconception health and more attention for social determinants of adverse pregnancy outcomes. This report lead to some changes over time with gradual establishment of so-called obstetric care collaborations which integrate primary and secondary care around hospitals, and local initiatives to link the welfare and health care system on a municipal level to provide optimal care for socially “vulnerable” mothers and their babies. However, we do not think that most changes have occurred at large scale before the end of our observation period.

Reviewer: 2

Reviewer Name: Hilary Brown

Institution and Country: University of Toronto

Please state any competing interests or state ‘None declared’:None declared.

Please leave your comments for the authors below

This study describes the impact of an urban perinatal health program in Rotterdam, Netherlands, on perinatal mortality, preterm birth, and small for gestational age. The study uses a difference-in-difference approach to test this natural experiment. The intervention includes many interesting and timely elements, including a focus on preconception health. The data are well-described, and the analysis is well-designed. My concerns relate to the design of the intervention itself and how suitable it is to evaluation using the authors' methods.

Major points:

- The authors state that the perinatal programme was introduced gradually across 2009 to 2012 (vs. a specific start date) and that not all of the components reached the entire city (e.g., primary care birth centres were only in certain neighbourhoods). Could the choice of looking at all (but 6) urban neighbourhoods in Rotterdam (vs. the specific neighbourhoods in which all of the components were implemented) explain why no effect was seen? It seems to me that the study population for this paper does not exactly match where the intervention was implemented, and therefore there is a mismatch between the study question and how it was tested.

Could the authors identify the neighbourhoods in which all elements of the program were implemented and compare these to the ones in which none were implemented?

Response: The intervention group consisted of all deliveries in all 51 urban neighbourhoods in 10 out of 14 boroughs in Rotterdam. We have general information on implementation of the key components of the perinatal programme at borough level, albeit not at the level of individual pregnant women. The following interventions showed mixed uptake across boroughs: peer education (5 boroughs), development of care pathways in midwifery practices and hospitals (3) and, lastly, the birth centre (4 boroughs).

We did analyses comparing neighbourhoods with high uptake on each intervention separately (three analyses) with neighbourhoods in the reference group, as in the main analysis. For none of these interventions any effect on the outcome measurers was observed (see table below).

As only 4 neighbourhoods had high uptake of 2 interventions, and none neighbourhoods of all 3 interventions, and the non-significant results, we refrained from further analyses on potential cumulative effects.

	perinatal mortality			SGA		preterm birth			
	OR	95% CI		beta	95% CI		OR	95% CI	
1. peer education	1.182 6	1.039 3	1.345 7	0.9825	0.948 9	1.017 1	0.963 4	0.9226	1.005 9
2. care pathways	1.116 5	0.951 2	1.310 4	0.9951	0.955 2	1.036 6	0.985 5	0.9364	1.037 2
3. birth centre	1.058 3	0.960 6	1.165 9	0.9896	0.705 4	1.013 9	0.970 6	0.9414	1.000 8

Sentence in our revised methods (page 8, paragraph 4):

“We conducted several sensitivity analyses for lagged programme effects, adjustment for covariate imbalance and the individual program component effects.... As a final sensitivity analysis, we studied a possible intervention effect in the boroughs that were targeted by these interventions, compared to the control population, using the same difference-in-difference model as in the main analysis.”

Sentence in our revised results (page 14, paragraph 2):

“No intervention effects were observed for individual programme components (supplementary file 8).”

- If the intervention was gradually introduced in 2009 to 2012, 2014 may not be a sufficient length of follow-up – i.e., depending on the component of the program there were only 2-5 years of post-intervention data. Why did the authors not analyze more recent data (e.g., up to 2018)?

Response: The Dutch perinatal registry requires about two years before the data is released due to

(partial) data collection on paper and data quality procedures. The dataset was the latest dataset when our research project started. Also, the municipality of Rotterdam participated in a sequel program to “Ready for a Baby”, Healthy Pregnancy for All, from 2012 onwards with similar objectives, so attribution of an effect in later years would be difficult (Vos, 2015).

- Only 43 couples attended an individual consultant session in all of Rotterdam. Although 2,300 individuals received peer counseling, the number of people who received the strongest “dose” of preconception care was very small. This should be considered more fully in the Discussion section.

Response: We included this in our revised discussion.

Paragraph in our revised discussion (page 18, paragraph 2):
“Only 43 couples attended an individual preconception consultation by a general practitioner in all of Rotterdam, which is too small to expect any effect on population level.”

Minor points:

- What are the medical and social risk factors responsible for the heightened adverse perinatal outcomes in deprived neighbourhoods in the Netherlands? An understanding of these factors would help to understand why the perinatal programme may not have made an impact (i.e., did it target these risk factors or something else?).

Response: The increased adverse perinatal outcomes in deprived neighbourhoods in the Netherlands can be attributed to an accumulation of sociodemographic (household income, migration background), lifestyle (maternal smoking), obstetric history (nullipara, complications previous pregnancy), and health-related risk factors (diabetes, hypertension) present within deprived neighbourhoods (Timmermans, 2011). Not all these risk factors are avoidable. Possibly avoidable risk factors include: smoking, alcohol and recreational drug use, BMI, pregnancy planning, folic acid use, gestational age at booking, maternal psychopathology, comorbidity, and STDs (Timmermans, 2011).

Both the preconception care campaign and peer education intervention as well as the Rotterdam Reproductive Risk Reduction (R4U) scorecard addressed a wide range of these avoidable risk factors, primarily health behaviours (Peters, 2014)(van Veen, 2015). Therefore, we think that the programme correctly targeted important risk factors, hence, the limited impact should be sought in the limited coverage of the programme as stated in the discussion (page 18, paragraph 2).

- What were the R4U risk factors that were targeted?

The R4U scorecard contains 77 items including medical risk factors like chronic disease, medication, previous pregnancies, and non-medical risk factors like substance abuse, domestic violence, low income, debt, psycho-social problems (van Veen, 2015). Some of these risks will be taken care by the health professional after identification (Vos, 2015). In the programme, 28 care pathways were defined to guide midwives and gynaecologists for appropriate referral to other medical and social professionals. Especially a systematic approach for the referral of non-medical risk factors to professionals in the social domain was new in the Dutch context. Unfortunately, although care pathways were identified during the Ready for a Baby programme, structured implementation and evaluation of implementation were not achieved during the programme.

FORMATTING AMENDMENTS (if any)

Required amendments will be listed here; please include these changes in your revised version:

1. Correct supplementary format:

- Please re-upload your Supplementary files in PDF format.

Done

2. Change file supplementary name:

- Kindly change the file of your Appendix 1 into Supplementary file 1 to avoid confusion.

Done

References

Peters IA, Schölerich VL, Veen DW, Steegers EA, Denктаş S. Reproductive health peer education for multicultural target groups. *Journal for Multicultural Education*. 2014, 8(3):162-78.

Soledad Cepeda M, Boston R, Farrar JT, Strom BL. Comparison of Logistic Regression versus Propensity Score When the Number of Events Is Low and There Are Multiple Confounders . *American Journal of Epidemiology*, 2003; 158(3):280–287.

Stuart EA. Matching Methods for Causal Inference: A Review and a Look Forward. *Statist. Sci.* Volume 25, Number 1 (2010), 1-21.

Timmermans S, Bonsel GJ, Steegers-Theunissen RP, Mackenbach JP, Steyerberg EW, Raat H, et al. Individual accumulation of heterogeneous risks explains perinatal inequalities within deprived neighbourhoods. *Eur J Epidemiol*. 2011 Feb;26(2):165-80.

Van Veen MJ, Birnie E, Poeran J, Torij HW, Steegers EA, Bonsel GJ. Feasibility and reliability of a newly developed antenatal risk score card in routine care. *Midwifery*. 2015 Jan;31(1):147-54.

Vos AA, van Voorst SF, Waelput AJ, de Jong-Potjer LC, Bonsel GJ, Steegers EA, Denктаş S. Effectiveness of score card-based antenatal risk selection, care pathways, and multidisciplinary consultation in the Healthy Pregnancy 4 All study (HP4ALL): study protocol for a cluster randomized controlled trial. *Trials*. 2015 Jan 6;16:8.

VERSION 2 – REVIEW

REVIEWER	Hilary Brown University of Toronto, Canada
REVIEW RETURNED	25-Sep-2019
GENERAL COMMENTS	The authors provide a thorough and thoughtful response to the reviews, including the addition of several helpful sensitivity analyses. I have no further comments.
GENERAL COMMENTS